# Capturing Household Structure and Mobility within and between Remote Aboriginal Communities in Northern Australia Using Longitudinal Data: A Pilot Study

**DOI:** 10.3390/ijerph191912002

**Published:** 2022-09-22

**Authors:** Jessie J. Goldsmith, Patricia T. Campbell, Juan Pablo Villanueva-Cabezas, Rebecca H. Chisholm, Melita McKinnon, George G. Gurruwiwi, Roslyn G. Dhurrkay, Alfred M. Dockery, Nicholas Geard, Steven Y. C. Tong, Jodie McVernon, Katherine B. Gibney

**Affiliations:** 1Department of Infectious Diseases, Peter Doherty Institute for Infection and Immunity, University of Melbourne, Melbourne, VIC 3000, Australia; 2Centre for Epidemiology and Biostatistics, Melbourne School of Population and Global Health, University of Melbourne, Parkville, VIC 3010, Australia; 3Department of Mathematical and Physical Sciences, La Trobe University, Bundoora, VIC 3086, Australia; 4Wellbeing and Preventable Chronic Diseases Division, Menzies School of Health Research, Charles Darwin University, Casuarina, NT 0811, Australia; 5Bankwest Curtin Economics Centre, Curtin University, Bentley, WA 6102, Australia; 6School of Computing and Information Systems, Faculty of Engineering and Information Technology, University of Melbourne, Parkville, VIC 3010, Australia; 7Victorian Infectious Diseases Service, Royal Melbourne Hospital, Peter Doherty Institute for Infection and Immunity, Parkville, VIC 3050, Australia; 8Victorian Infectious Diseases Reference Laboratory, Royal Melbourne Hospital, Peter Doherty Institute for Infection and Immunity, Melbourne, VIC 3000, Australia

**Keywords:** aboriginal, indigenous, contact patterns, household structure, disease transmission, household model, human mobility

## Abstract

Cultural practices and development level can influence a population’s household structures and mixing patterns. Within some populations, households can be organized across multiple dwellings. This likely affects the spread of infectious disease through these communities; however, current demographic data collection tools do not record these data. Methods: Between June and October 2018, the Contact And Mobility Patterns in remote Aboriginal Australian communities (CAMP-remote) pilot study recruited Aboriginal mothers with infants in a remote northern Australian community to complete a monthly iPad-based contact survey. Results: Thirteen mother–infant pairs (participants) completed 69 study visits between recruitment and the end of May 2019. Participants reported they and their other children slept in 28 dwellings during the study. The median dwelling occupancy, defined as people sleeping in the same dwelling on the previous night, was ten (range: 3.5–25). Participants who completed at least three responses (n = 8) slept in a median of three dwellings (range: 2–9). Each month, a median of 28% (range: 0–63%) of the participants travelled out of the community. Including these data in disease transmission models amplified estimates of infectious disease spread in the study community, compared to models parameterized using census data. Conclusions: The lack of data on mixing patterns in populations where households can be organized across dwellings may impact the accuracy of infectious disease models for these communities and the efficacy of public health actions they inform.

## 1. Introduction

The pattern of mixing between people in a population is a crucial determinant of how infectious diseases spread within it [1]. As mathematical models exploring infection dynamics gain traction as tools to guide infectious disease prevention and control activities in different populations, the adequate quantification of mixing patterns has become critical. Across the globe, the social and cultural practices of different populations result in different mixing patterns [2,3,4,5,6,7,8]. In Australia, the social and cultural practices of remote-living Aboriginal and Torres Strait Islander people (hereafter Aboriginal Australians) result in unique mixing patterns [2,3,4,8]. These mixing patterns may influence how infectious diseases spread through remote Aboriginal communities [9].

Community mixing patterns can be conceptualized in three parts: household mixing, intra-community mixing between people from different households, and temporary mobility into and out of the community. 

First, households can be pivotal in the spread of infectious disease [10]. For example, the attack rate for SARS-CoV-2 is more than five times higher among household contacts than non-household contacts [11]. Australian demographic surveys show that, on average, dwellings in remote Aboriginal communities have more occupants than dwellings in either urban Aboriginal communities or non-Aboriginal communities [12,13]. These surveys typically assume that a household consists of a single dwelling. However, in remote Aboriginal communities, households may be organized across multiple dwellings [8,9]. To understand true household mixing relevant to transmission of pathogens, we may need to consider connections between dwellings. To date, only one study has quantified these linkages [2]. It is based on the frequency that individuals stayed in a single dwelling in a remote Aboriginal Australian community over the course of a year [2]. Using data from this study, Chisholm et al. developed an infectious diseases model that compared outbreak dynamics for populations with multi-dwelling households versus single dwelling households; the modelled outbreak scenarios with multi-dwelling households had larger and faster outbreaks [9]. Clearly, more data are needed to support this type of analysis.

Second, intra-community mixing patterns can also contribute to the spread of infectious disease [14]. Remote-living Aboriginal Australian adults are more frequently involved in cultural events, ceremonies and other community events, have more frequent face to face contact with people outside of the home and are more likely to provide support to relatives outside the household than non-remote-living Aboriginal Australians [3]. While surveys to collect intra-community mixing patterns are increasingly being used in other settings [15,16,17,18], there are no comparable studies for remote-living Aboriginal Australians [9].

Finally, temporary mobility into and out of a community provides opportunities for the introduction or exportation of pathogens. Temporary mobility can be defined as a person being absent from their usual place of residence for up to six months [19]. This is consistent with the definition of ‘usual residence’ used by the Australian census [20]. Disease transmission models that include temporary mobility typically rely on Australian Census data for parameterization [21]. Census data indicate Aboriginal Australians have higher rates of temporary mobility than Australians living in urban areas and non-Aboriginal Australians living in rural and remote areas [4]. These census data, however, do not collect information on dynamic patterns of temporary mobility or linkages between communities.

Current demographic data collection tools do not reflect the cultural and societal norms of remote-living Aboriginal Australians [22,23]. Additional data are required to characterize mixing patterns in these communities. The Contact And Mobility Patterns in remote Aboriginal Australian communities (CAMP-remote) pilot study was developed to begin addressing this knowledge gap by quantifying household mixing (including multi-dwelling households), intra-community social contact patterns and temporary mobility patterns for a remote Aboriginal Australian community. In this paper, we describe the CAMP-remote pilot study, analyse the data collected, and demonstrate its application in a mathematical model that simulates the transmission of an infectious disease within the study community. We discuss the impact of using these data to enhance our understanding of mixing patterns in remote Aboriginal Australian communities.

## 2. Materials and Methods

The project protocol and data collection tools were developed and refined during consultation with community members and stakeholders from June 2017 to April 2018. These consultations included researchers, healthcare workers and board members from the study community, as well as staff from the University of Melbourne, Curtin University, Menzies School of Health Research, One Disease, and Miwatj Health Aboriginal Corporation. This study was approved by the Human Research Ethics Committee of Northern Territory Department of Health and Menzies School of Health Research (HREC 2017-2917), the Community Board, and the Miwatj Health Aboriginal Corporation.

### 2.1. Study Area and Study Design

The study was conducted between June 2018 and May 2019 in a remote Aboriginal community in the Northern Territory, Australia. The community population is between 2000 and 2500 and it is approximately 1.5 h flying time from Darwin, the capital of the Northern Territory. The largest employers within the community are supermarkets.

#### 2.1.1. Study Participants

Study participants had to live in the study community and identify as female, Aboriginal and the primary carer of an infant no more than one year old. Potential participants were excluded if they were unable to give informed consent or if they usually shared a dwelling with another mother–infant pair who was already enrolled in the study.

The decision to enrol mother–infant pairs in this pilot study was made during the consultation process. This cohort was assumed to be less likely to be lost to follow-up and provided an opportunity to capture social interactions across a spectrum of ages within the community. 

#### 2.1.2. Study Procedures

Two survey instruments were used. Both were conducted in person by study staff. The baseline survey was undertaken at enrolment and gathered basic demographic information about the participant and any children for whom they were the primary caregiver. The second was the dwelling, contact and travel survey which was repeated monthly from enrolment until May 2019.

Survey questionnaires were based on previously validated surveys (Appendix A) [15,24]. The contact questions were based on the questionnaire used in the European POLYMOD study [15], previously adapted for use in urban Australian populations [25], while the temporary mobility questions were based on the Mobility Survey conducted in Central Australia by the Co-operative Research Centre for Remote Economic Participation [24]. Participants were asked how many people they had contact with in the 24 h prior to the survey. Compared to previous surveys, questions on contacts were simplified to reduce the reporting burden on participants. Contacts were categorized by age-group (preschool [less than 5 years], school [5–14 years] and adult [15+ years]) and participants only reported contacts that met the following criteria:Dwelling contact—anyone who slept in the same dwelling as the participant the previous night.Social contact—anyone with whom the participant spent an hour or more during the previous 24 h, excluding dwelling contacts.

Survey responses were collected using a custom-built iPad application. The questions were written in English, but Aboriginal community researchers were able to conduct the survey in the local language, if required. The iPad application made use of visual representation rather than text wherever possible, following feedback during the consultation process. For example, a map of the community was used to collect information about dwelling location.

### 2.2. Data Analysis

All analyses were undertaken using the statistical software R [26]. Graphs were produced using ggplot2 and nptest was used to generate non-parametric bootstrapped confidence intervals [27,28].

#### 2.2.1. Clustering Responses by Dwelling Location

The dwellings in which participants indicated they and/or their children slept over the week prior to each survey were recorded as spatial co-ordinates. A hierarchical clustering approach was used to group the spatial coordinates and identify distinct dwellings [26]. Dwellings were defined as the centroid of 50 m radius buffers to account for variation in the coordinates resulting from a touch screen application. Spatial data were mapped using the Australian Albers project (epsg projection 3577—gda99).

The median number of dwellings slept at by participants and their children and the proportion of surveyed nights spent at each dwelling were calculated using data from participants who responded to the monthly survey at least three times. Dwellings were classed as ‘core’, ‘frequent’ and ‘infrequent’ according to the amount of time participants spent there, where core dwellings were the dwellings that participants reported staying at most regularly. The proportion of time spent at each class of dwelling was reported with 95% confidence intervals.

#### 2.2.2. Dwelling Contact Matrices

The repeated surveys collected dwelling occupancy for the night prior to each survey. The median and range of the mean occupancy was calculated per available room in each dwelling and within the room where the participant slept. In addition, the median and range of the mean occupancy per dwelling was calculated by age group. These data were presented as age-stratified matrices with 95% confidence intervals (Appendix B).

#### 2.2.3. Intra-Community Mixing

The number of social contacts who spent an hour or more with the participant in the 24 h period leading up to each study visit was captured in the repeated surveys, excluding dwelling contacts. The mean number of daily social contacts per participant was reported.

#### 2.2.4. Temporary Mobility

The repeated surveys collected data on travel by participants out of the community in the preceding month, including the primary reason for and duration of travel, and whether their baby travelled with the participant. In addition, researchers noted if a participant was out of the community during one of the monthly study visits. If a participant was absent on two successive survey visits, it was assumed they had not returned to the community in the intervening period.

#### 2.2.5. Individual Based Model Simulation

We compared the results of simulations for an influenza-like illness and an endemic infection using an individual-based transmission model which was informed by different assumptions and data sources for community mixing. The purpose was to quantify the impact of different assumptions of household structure and community mixing on model outputs and estimate the importance of collecting multi-dwelling household structure and temporary mobility data for model accuracy in remote Aboriginal communities [9]. For each disease type, we simulated the model using three scenarios for a community with the demographic characteristics of our study community. First, we assumed that each person is equally likely to contact each other person in the community (homogenous mixing). Second, we used contact matrices developed using Census data from the Australian Bureau of Statistics on household size and age distribution, under the assumption that each household is associated with a unique dwelling [29]. Third, we used the dwelling contact matrices, multi-dwelling household structure and temporary mobility rates developed using the CAMP-remote data. Social contact data for the second and third scenarios were sourced from Kiti et al.’s study based in Kenya [16]. For the influenza-like illness, we compared the timing and magnitude of the infection peak for the three scenarios. For the endemic disease, we compared the disease prevalence between the three scenarios.

Further detail on the model structure and parameters is in Appendix C.

## 3. Results

### 3.1. Summary Statistics

Thirteen mother–infant pairs were recruited: ten in June 2018, two in August 2018 and one in October 2018. Three participants withdrew over the course of the study, resulting in a 23% loss to follow-up. Sixty-nine survey responses were collected, a median of four responses per participant (IQR: 2–9). Further detail on recruitment and participation is provided in Appendix D.

At recruitment, the median age of participants was 23 years (range: 17–37 years) and infants was eight months (range: 0–22 months). Participants reported caring for a median of two children (range: 1–3).

### 3.2. Household Structures and Size

We identified 28 distinct dwellings where a participant and her infant or another of the children she cared for had slept during the week prior to a completed survey. Participants who responded to the survey at least three times (n = 8) identified a median of three separate dwellings (range: 2–9, Figure 1).

Of the 28 dwellings identified, the composition on the night prior to the survey was collected on at least one occasion for 25 dwellings and collected five or more times for four of the dwellings. The mean occupancy by dwelling ranged from 3.5 to 25 people, with the median of the mean of ten people (IQR: 8–12, Figure 2). Most of these occupants were adults.

A comparison of the dwellings where five or more nights of occupancy data were available (Figure 3) demonstrates significant variation in the number of occupants for some dwellings (e.g., Dwelling D) but not others (e.g., Dwelling A).

The median of the mean occupancy per room by dwelling was 2.7 individuals (range: 1.6–4.2), with a higher median of the mean occupancy in the participant’s room of 3.3 individuals (range 2.0–5.0). Participants reported always sleeping in the same room as their baby and almost always (97% of occasions surveyed) in the same bed.

### 3.3. Dwelling Contacts

We estimate this population had 15 adult-to-adult (95% CI: 10–28) and 12 adult-to-young (95% CI: 6–16) dwelling contacts each day (Figure 4). 

### 3.4. Intra-Community Mixing

The median of the mean number of social contacts (that were not dwelling contacts) reported per participant for a day was two (range: 0–11), Figure 5. 

### 3.5. Temporary Mobility

A median of 28% (range: 0–63%) of participants travelled out of the community each month. No visual evidence of seasonality was apparent between the wet (November to April) and dry (May to October) seasons (Appendix E).

From the subset of surveys in which participants reported travel (n = 22), it was usually within the region where the study community was located (n = 13, 59%) and lasted for a median of seven days (IQR: 5–18 days). The primary reason for travel was most frequently to attend a funeral (n = 9, 41%) or visit family (n = 7, 32%).

### 3.6. Individual Based Model Simulation

The household structured model parameterized with the CAMP-remote data had the highest and earliest peak of infectious individuals in the simulations of an outbreak (Figure 6a) and the highest on-going prevalence in the simulations of an endemic infection (Figure 6b). Applying the ABS data resulted in simulated outbreaks and endemic transmission that were overall similar to those for the models parameterized with the CAMP-remote data, but with a slightly lower and later peak and a slightly longer duration for the outbreak model, and slightly lower on-going prevalence for the endemic disease model. Uniform mixing assumptions resulted in simulated outbreaks with a substantially lower and later peak and the simulated endemic transmission with substantially lower on-going prevalence (Appendix F).

## 4. Discussion

We documented that participants and their children slept in a median of three dwellings over the 12-month study period and more than a quarter of participants travelled out of the community each month. As we only have data on where a participant and her children slept for a small fraction of the year, and it was assumed that participants did not return to the community between surveys if they were absent for successive surveys, these estimates likely underestimate the extent of multi-dwelling households and temporary mobility in this remote Aboriginal community. Even using these lower-bound estimates, modelling has demonstrated the importance of characterizing these data for model accuracy and to subsequently inform infectious disease prevention and intervention strategies.

These socio-cultural patterns influence how infectious diseases are transmitted in a community and are an important consideration when planning and implementing a public health response. A recent systematic review found that most prior social contact surveys have been conducted in high-income countries [30]. Within Oceania, the review found that Australia was the only country that has collected data on mixing patterns [30]. Cultural norms, development level and geography influence social mixing, household size and structure [30]. For example, Vino et al. demonstrated that household contact patterns in remote-living Australian Aboriginal communities were substantially higher than those for residents of Melbourne, Australia [12].

Fluid households and substantial rates of temporary mobility are not unique to remote-living Aboriginal Australians. Similar patterns have been observed in South Africa, Tanzania, Burkina Faso and in remote Canadian Inuit communities [5,6,7]. In recognition of the fluidity of household membership within South Africa, the Population Intervention Platform (PIP) cohort, which was established in 2000 (African Centre Demographic Information System cohort) in rural KwaZulu Natal, uses a definition of household membership that includes resident and non-resident members and allows for individuals to belong to more than one household [31,32]. Further studies are required that collect information on contact patterns within communities exhibiting these patterns. The CAMP-remote pilot study provides a demonstration of both a method for and the impact of collecting these data.

Our findings are consistent with those previously reported by Musharbash [2]. Musharbash’s study collected data on the number of people that stayed at a single dwelling in a remote community in Central Australia over 221 nights and, based on these data, proposed four categories of occupant: core residents (those present between 60–100% of the time); regular residents (those present between 20–34% of the time); other residents (present 4–16% of the time); and sporadic short-term visitors [2]. While the consistency between these two studies is encouraging, further data are required to confirm the findings. The repeated survey approach of the CAMP-remote study yielded valuable data on connections between dwellings and variation in dwelling occupancy that otherwise would not have been available.

The dwelling contact matrix developed in the CAMP-remote study estimated a median of 12 adult-to-young child contacts each day per dwelling, compared with the mean of 4.14 per household for remote Australian Aboriginal communities reported in Vino et al., based on data from the Aboriginal Birth Cohort study [12]. While dwellings in the CAMP-remote study all included at least one mother and one infant (due to the recruitment strategy), dwellings in the Aboriginal Birth Cohort study all included at least one individual aged between 22 and 27 years [12]. Differences in the contact matrix calculated using the CAMP-remote data and the contact matrix in Vino et al. may be a function of the different study cohorts or differences in the age group categories, communities, study sample sizes, or study designs.

The CAMP-remote survey is the first study to collect quantitative data on intra-community mixing within a remote Aboriginal community. Direct comparisons between data collected during this study and prior studies are challenging due to differences in definitions for social contact. Kiti et al.’s study, based in Kenya, defines a social contact as anyone with whom the participant had physical contact [16]. The POLYMOD study from Europe similarly defines a social contact as anyone with whom the participant had skin-to-skin contact or an in-person two-way conversation with three or more words without skin-to-skin contact [15]. By comparison, the definition used in the CAMP-remote study was anyone with whom the participant spent an hour or more. However, when mean contacts observed in the CAMP-remote pilot study are contrasted to that of similarly aged women (20–49 years) in Kiti et al.’s study, we observe that while the scales differ, the distributions of the number of social contacts per day are similar (Appendix G). As Kiti et al. collected social contact information from all aged groups and both sexes, using Kiti et al.’s data for our simulations remains a reasonable substitute for data on intra-community mixing patterns within remote-living Aboriginal communities until more comprehensive social contact data become available for remote Aboriginal Australian communities.

The definition of social contact adopted during the CAMP-remote study was developed in consultation with the community and designed to both limit reporting burden for participants and maximize the chances of on-going participation in the study. While this approach was appropriate given the longitudinal nature of the CAMP-remote study, it limited direct comparison of social contact data with previously published studies. An alternative approach would be to collect social contact data from a remote Aboriginal community using a previously validated definition (e.g., from the POLYMOD survey or study by Kiti et al.) on a single occasion to allow comparison with other, disparate, populations.

The modelling results demonstrate that including information on household characteristics and temporary mobility amplifies estimates of the spread of an infectious disease in remote Aboriginal communities. Differences were observed in the intensity of transmission between the three scenarios, with the modelling results from the simulation using CAMP-remote data having the most intense outbreaks and highest rates of endemic transmission. Both models that applied a household structure showed substantially greater intensity of transmission than the model that assumed uniform mixing, suggesting that uniform mixing assumptions are not appropriate to parametrize infection dynamics within these communities. Prior work by Chisholm et al. suggests that we would expect to observe an even greater difference in transmission intensity between the two household-structured models (parameterized by ABS versus CAMP remote data) if we were to consider a less-transmissible pathogen [9], indicating the increased importance of collecting relevant mobility, social contact, and household data for such pathogens.

We note that Frances Morphy has argued that a nodal network better captures the structure of remote Aboriginal communities than the conceptualization of households [22]. However, the quality and quantity of contact between individuals that share living spaces differs from those who do not, which is important when modelling transmission of infectious disease, and we have taken care to differentiate individual dwellings from households.

Our study has several limitations. First, the small sample size means we are unable to determine the statistical significance of our results. Second, as the CAMP-remote study only recruited mother–infant pairs, data are limited to this cohort and may not be generalizable to all members of remote Aboriginal communities. To represent the household structure, intra-community mixing patterns and temporary mobility of remote-living Aboriginal Australians more accurately further, more inclusive data collection is required. Third, the extent of loss to follow-up and missing survey responses may have introduced bias if the participants who dropped out or did not respond were different from those who did respond. Finally, to identify individual dwellings it was necessary to make assumptions about the distance between dwellings which may have resulted in dwellings being misclassified (i.e., adjacent dwelling to be classified as a single dwelling).

## 5. Conclusions

Our study demonstrates the gap in the current knowledge of mixing patterns within the remote Aboriginal Australian communities and the importance of collecting data that reflect the organization of households across dwellings to inform infectious disease modelling and public health interventions for these communities and others with similar patterns. Given the limited proximate health infrastructure in these settings it is vital that we develop models that are as accurate as possible to allow appropriate planning for, prevention of, and responses to an outbreak. Similarly, public health responses to endemic disease need to be based on a realistic understanding of the extent of transmission and the level of community participation required to make a meaningful difference in the health outcomes of individuals. The types of data that the CAMP-remote pilot study demonstrated could be collected contributes to this goal.

## Figures and Tables

**Figure 1 ijerph-19-12002-f001:**
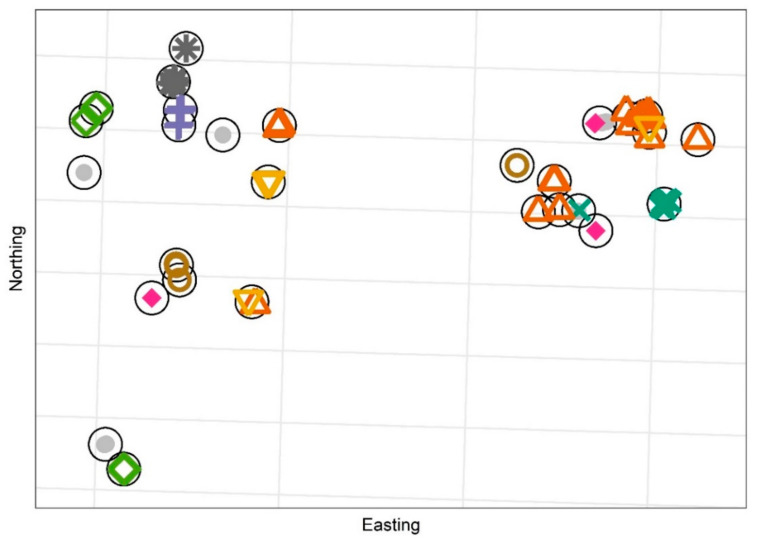
Clustering of spatial coordinates to identify distinct dwellings (n = 28). Each symbol represents a survey response. Participants with one or two responses are depicted as grey circles, all other participants (n = 8) are depicted using a unique symbol.

**Figure 2 ijerph-19-12002-f002:**
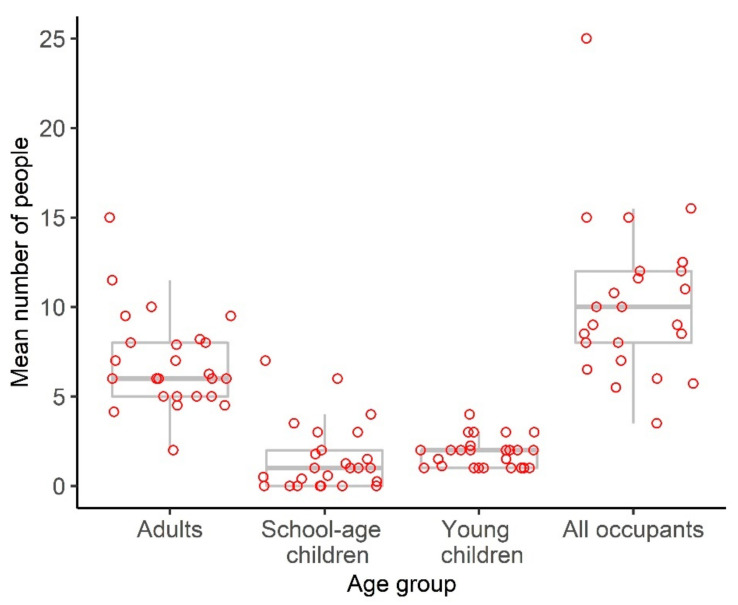
Distribution of the mean number of people per dwelling by age group. Each dot represents the mean number of individuals of a specific age across all survey responses for a given dwelling. The median of the means is represented using the thick grey line and the interquartile range of the means with the thinner grey lines.

**Figure 3 ijerph-19-12002-f003:**
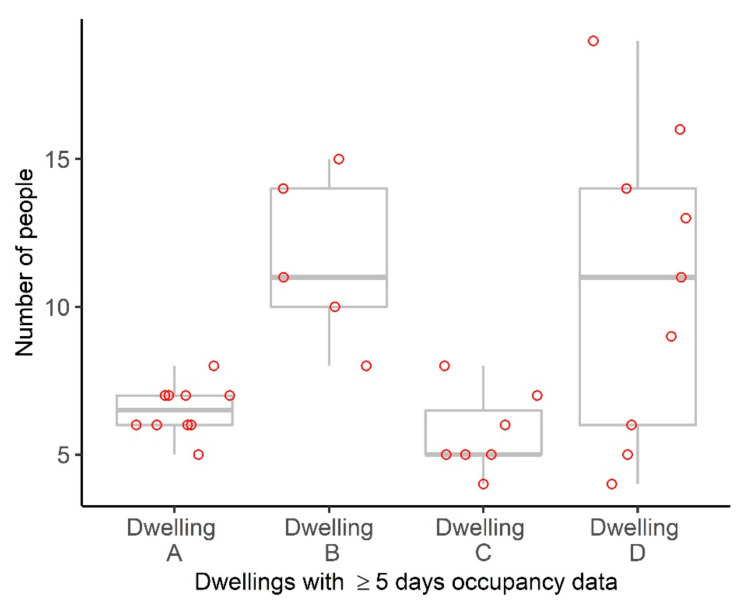
Distribution of the number of people per dwelling for dwellings with five or more nights of data. Each dot represents the total number of individuals in each dwelling for a specific survey response. The median is represented using the thick grey line and the interquartile range with the thinner grey lines.

**Figure 4 ijerph-19-12002-f004:**
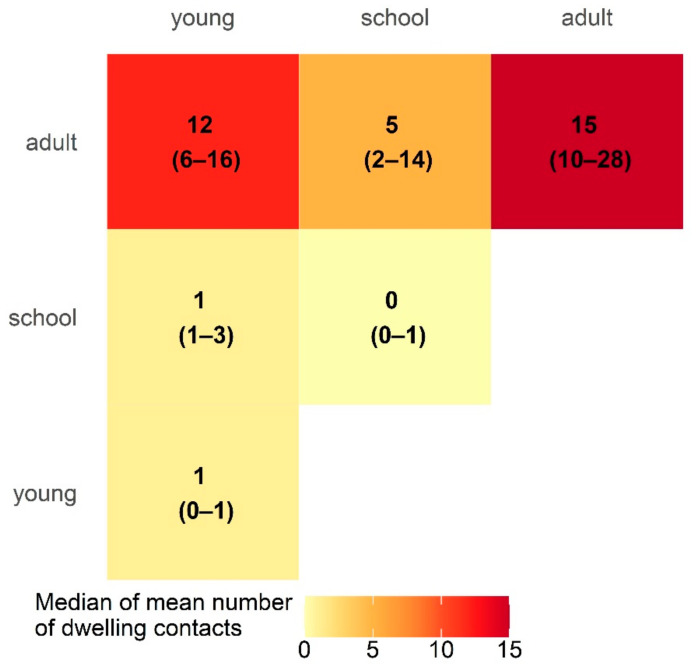
Median number of mean dwelling contacts between each age category per dwelling using the CAMP-remote data. Confidence intervals of 95%, estimated with a nonparametric bootstrap method, are indicated in brackets. Age ranges were young (less than 5 years), school-aged (5–14 years) and adult (15+ years).

**Figure 5 ijerph-19-12002-f005:**
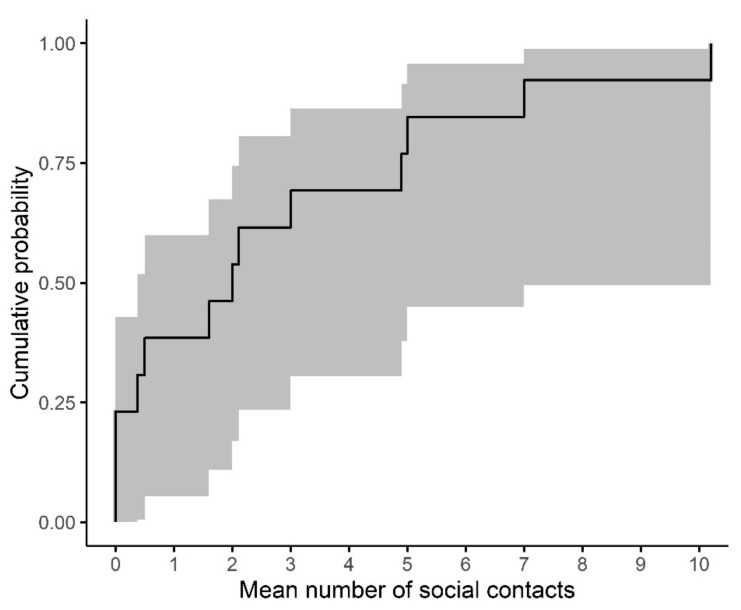
Cumulative distribution of the mean number of social contacts per CAMP-remote participant (n = 13) per day. Shaded area represents 95% confidence interval.

**Figure 6 ijerph-19-12002-f006:**
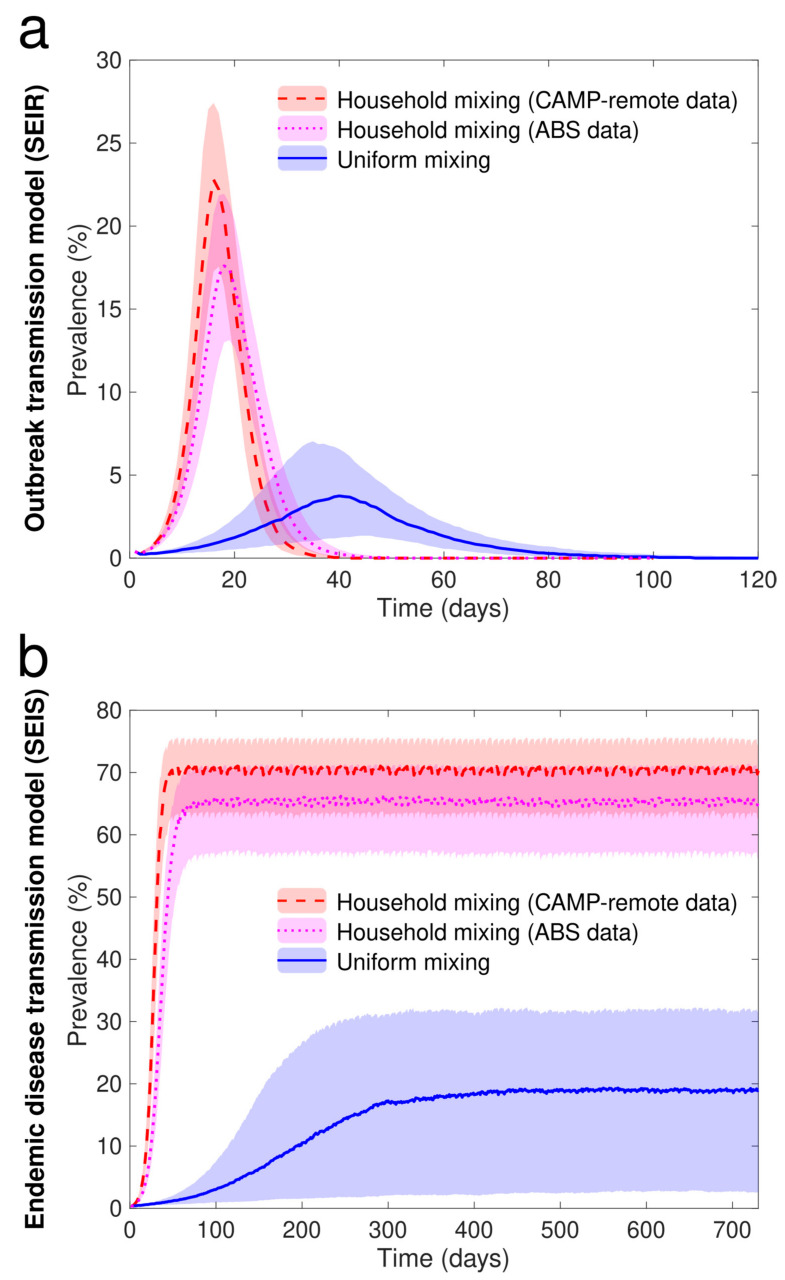
(**a**) The prevalence of infection arising from simulations of an influenza-like SEIR model under different mixing assumptions. (**b**) The prevalence of infection arising from simulations of an endemic disease in an SEIS model under different mixing assumptions. Both models are parameterized to reflect the study community.

## Data Availability

The data presented in this study are available on request from the corresponding author. The data are not publicly available because they contain information that could compromise the privacy of research participants.

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
