# Peer review of "Capturing Household Structure and Mobility within and between Remote Aboriginal Communities in Northern Australia Using Longitudinal Data: A Pilot Study"

_ijerph, 2022, doi:10.3390/ijerph191912002_

Round 1
Reviewer 1 Report
I would like to commend the authors on this interesting study that explored contact and mobility patterns in remote Aboriginal communities in Northern Territory. Overall, this study is clear and extremely well written. We often hear anecdotal evidence on the mobility within Aboriginal communities and this study has captured household, community social contact and temporary mobility well, even if it is within a small sample size. This data is particularly important in the current context of the COVID-19 Pandemic.
I have a few minor comments for the authors.
Abstract
I suggest the authors delete the word ‘negatively’ from the last sentence. The findings of this study definitely do impact the accuracy of infectious disease models for remote Aboriginal communities and the efficacy of public health actions, however, the word ‘negative’ is very harsh. Please also note the word negatively is not used again within the manuscript.
Introduction
The introduction clearly sets out the context as to why this study is important.
Methods
I commend the authors on their level of community consultation and adapting the survey tools based on community consultation for example using the map.
Results
The results were presented clearly. In section 3.4 I found the sentence particularly confusing – perhaps the authors can clarify this sentence.
Discussion
The discussion summarises this study well. It appropriately discusses the implications of this study that policy makers must take these findings in to consideration when disease modelling and planning public health responses and policies to effectively contain the spread of disease, particularly in vulnerable populations and when healthcare infrastructure is limited. This may be out of the scope of this paper, however the authors may want to consider how taking community movement and dwelling fluidity into consideration for policy makers is integral to the cultural safety for Aboriginal people when planning public health responses in disease outbreaks.
Author Response
We would like to thank the reviewer for taking the time to review our manuscript and providing useful suggestions. In response to their comments, we propose:
Comment 1: I suggest the authors delete the word ‘negatively’ from the last sentence. The findings of this study definitely do impact the accuracy of infectious disease models for remote Aboriginal communities and the efficacy of public health actions, however, the word ‘negative’ is very harsh. Please also note the word negatively is not used again within the manuscript.
Response to comment 1: We have removed the word ‘negative’ from the abstract
Comment 2: In section 3.4 I found the sentence particularly confusing – perhaps the authors can clarify this sentence.
Response to comment 2: We have reworded section 3.4 (lines 217-273) for additional clarity as follows:
“The median of the mean number of social contacts (that were not dwelling contacts) reported per participant for a day was two (range: 0–11), Figure 5. “
Comment 3: This may be out of the scope of this paper, however the authors may want to consider how taking community movement and dwelling fluidity into consideration for policy makers is integral to the cultural safety for Aboriginal people when planning public health responses in disease outbreaks.
Response to comment 3: The reviewer’s comment on the importance of considering the unique contact patterns of Aboriginal people when planning public health responses to ensure that they are culturally safe is an important point and perhaps this work could support further research on this topic in the future. However, it is outside the scope of the current study.
Reviewer 2 Report
A sound paper, but one that overlooks a significant body of work undertaken by Frances Morphy, one of Australia's few anthropologists with a keen interest in demography. Much of her work focussed on Yolgnu people so is directly relevant to this study. I think the analysis would be sharpened by reference to some of her papers:
1. On how the ABS conceptualises 'households' - Morphy, Frances. “UNCONTAINED SUBJECTS: ‘POPULATION’ AND ‘HOUSEHOLD’ IN REMOTE ABORIGINAL AUSTRALIA.” Journal of Population Research, vol. 24, no. 2, 2007, pp. 163–84. JSTOR, http://www.jstor.org/stable/41110887.
2. On mobility and the census in ArnhemLand - Mobility and its consequences: the 2006 enumeration in the north-east Arnhem Land region (PDF, 779KB), in Morphy (2006)(ed) Agency, Contingency and Census Process: Observations of the 2006 Indigenous Enumeration Strategy in remote Aboriginal Australia.
And 3. Morphy, Frances. Population, People and Place: The Fitzroy Valley Population Project. CAEPR WORKING PAPER No. 70/2010 (CAEPRWP70_0.pdf (anu.edu.au)) - this paper builds on Musharbash's conceptualisations of different levels or layering of residence.
While the paper contains reflects sound methodology, the conceptualisation of 'household' as the basis for family/sociality is arguably ethnocentric and perhaps not the best starting point for any genuine inquiry that seeks to capture mobility between and within communities. Having Indigenous people from the target community - presume Maningrida - guide the research approach would have honed its usefulness enormously.
Author Response
We would like to thank the reviewer for taking the time to review our manuscript and drawing our attention to Frances Morphy’s work. In response to their comments, we propose:
Comment 1: A sound paper, but one that overlooks a significant body of work undertaken by Frances Morphy, one of Australia's few anthropologists with a keen interest in demography. Much of her work focussed on Yolgnu people so is directly relevant to this study. I think the analysis would be sharpened by reference to some of her papers:
- On how the ABS conceptualises 'households' - Morphy, Frances. “UNCONTAINED SUBJECTS: ‘POPULATION’ AND ‘HOUSEHOLD’ IN REMOTE ABORIGINAL AUSTRALIA.” Journal of Population Research, vol. 24, no. 2, 2007, pp. 163–84. JSTOR, http://www.jstor.org/stable/41110887.
- On mobility and the census in Arnhem Land - Mobility and its consequences: the 2006 enumeration in the north-east Arnhem Land region (PDF, 779KB), in Morphy (2006)(ed) Agency, Contingency and Census Process: Observations of the 2006 Indigenous Enumeration Strategy in remote Aboriginal Australia.
And 3. Morphy, Frances. Population, People and Place: The Fitzroy Valley Population Project. CAEPR WORKING PAPER No. 70/2010 (CAEPRWP70_0.pdf (anu.edu.au)) - this paper builds on Musharbash's conceptualisations of different levels or layering of residence.
Response to comment 1: Adding as references to the following sentence on lines 106 and 107 of the manuscript: “Current demographic data collection tools do not reflect the cultural and societal norms of remote-living Aboriginal Australians.”
- Morphy F. Uncontained subjects: Population and household in remote aboriginal Australia. J Popul Res. 2007;24(2):163-84. doi: 10.1007/BF03031929.
- Morphy F. Mobility and its consequences: the 2006 enumeration in the north-east Arnhem Land region. In: Morphy F, editor. Agency, Contingency and Census Process: Observations of the 2006 Indigenous Enumeration Strategy in Remote Aboriginal Australia. Centre for Aboriginal Economic Policy Research 2007
Adding as a reference to the following sentence on line 78 and 79 of the manuscript: “However, in remote Aboriginal communities, households may be organized across multiple dwellings.”
- Morphy F. Population, people and place: the Fitzroy Valley population project. Canberra, Australia: Centre for Aboriginal Economic Policy Research, The Australian National University; 2018.
And to the following sentences on lines 65 to 68: “Across the globe, the social and cultural practices of different populations result in different mixing patterns [2-7]. In Australia, the social and cultural practices of remote-living Aboriginal and Torres Strait Islander people (hereafter Aboriginal Australians) result in unique mixing patterns [2-4].”
Comment 2: While the paper contains reflects sound methodology, the conceptualisation of 'household' as the basis for family/sociality is arguably ethnocentric and perhaps not the best starting point for any genuine inquiry that seeks to capture mobility between and within communities. Having Indigenous people from the target community - presume Maningrida - guide the research approach would have honed its usefulness enormously.
Response to comment 2: We agree that the conceptualisation of a ‘household’ is different in different populations. In populations where ‘household’ and ‘house’ are interchangeable, we note that from an infectious disease perspective the quality and quantity of contact between individuals within the same household differs from individuals that are not part of their household. How the contact differs is population specific but that the contact differs is universal. This has implications for the transmission of infectious disease and modelling must reflect that. For this study, our analyses focused on the dwellings as these have the greatest implication for the transmission of infectious diseases, rather than households/family groups which may be spread across multiple dwellings.
We further note, as stated in the paper lines 118 -123, extensive consultation was undertaken, during the development of the project protocol and survey instruments. This included study community members.
Therefore, we have added the following to the discussion after line 388:
“We note that Frances Morphy has argued that a nodal network better captures the structure of remote Aboriginal communities than the conceptualization of households [22]. However, the quality and quantity of contact between individuals that share living spaces differs from those who do not, which is important when modelling transmission of infectious disease, and we have taken care to differentiate individual dwellings from households.”